# Clinical Impact of Treating Versus Not Treating Asymptomatic Bacteriuria/Candiduria in the First Two Months After Kidney Transplantation

**DOI:** 10.3390/antibiotics14111155

**Published:** 2025-11-14

**Authors:** Biagio Pinchera, Rosa Carrano, Isabella Di Filippo, Vincenzo Fotticchia, Mariangela Petrone, Francesco Antimo Alfè, Gianmarco Borriello, Amerigo Piccione, Fabrizio Salemi, Ivan Gentile

**Affiliations:** 1Section of Infectious Diseases, Department of Clinical Medicine and Surgery, University of Naples “Federico II”, 80131 Naples, Italy; vincenzo.fotticchia@unina.it (V.F.);; 2Section of Nephrology, Department of Public Health, University of Naples “Federico II”, 80131 Naples, Italy; 3Section of Infectious Diseases, Presidio Ospedaliero “A. Cardarelli”, 86100 Campobasso, Italy; isadifi93@gmail.com

**Keywords:** asymptomatic bacteriuria, asymptomatic candiduria, kidney transplantation, treatment

## Abstract

**Background/Objectives:** The management of asymptomatic bacteriuria (ASB) and candiduria (ASC) in kidney transplant recipients during the early post-transplant period is controversial. This study aimed to evaluate whether treating, versus not treating, ASB and ASC episodes in the first two months after kidney transplantation influences clinical outcomes and the emergence of multidrug-resistant (MDR) infections. **Methods:** We conducted a single-center retrospective cohort study enrolling patients with ASB or ASC occurring in the first two months after kidney transplantation between January 2019 and July 2024. Patients were classified into treated and untreated groups. The primary endpoint was 30-day mortality. Secondary endpoints included mortality at 90, 180 and 360 days; incidence of sepsis or septic shock; bacteremia/candidemia, hospitalization, graft loss; decline in renal function, urinary tract infections (UTIs), recurrent UTI and rate of MDR colonization/infection. **Results:** We enrolled 59 kidney transplant recipients and observed 147 episodes of ASB/ASC. Of the 147 episodes, 95 were untreated and 52 were treated. No significant differences were observed between treated and untreated patients in 30-day (2.1% vs. 3.8%) or 90-day mortality (2.1% vs. 1.9%), nor in any of the secondary clinical outcomes. However, patients who received treatment tended to have a higher rate of MDR colonization/infection (63% vs. 46%). MDR pathogen isolation was significantly associated with increased risks of septic shock (OR 4.639, *p* = 0.04), bacteremia/candidemia (OR 3.734, *p* = 0.01), hospitalization (OR 2.183, *p* = 0.03) and renal function deterioration (OR 3.93, *p* = 0.03). **Conclusions:** Antimicrobial treatment of ASB and ASC in the early post-transplant period would seem not to confer clinical benefit and may be associated with the risk of MDR colonization/infection.

## 1. Introduction

Management of asymptomatic bacteriuria (ASB) and asymptomatic candiduria (AC) during the first two months after kidney transplantation remains contentious. The chief concern is that these colonizations might progress to symptomatic infection, compromise graft function, or expose recipients to unnecessary antimicrobials. Yet the evidence supporting treatment in the absence of symptoms is both scarce and inconsistent; therefore, current international guidelines recommend a cautious, individualized approach [1,2,3].

Historically, any positive urine culture in a transplant recipient triggered empirical therapy, motivated by fears that untreated ASB could herald severe complications. Emerging data, however, suggest that routine treatment may be counter-productive: it can foster multidrug-resistant (MDR) organisms, disrupt urinary microbiota, increase the incidence of candiduria, and may ultimately impair graft outcomes [4,5].

The 2019 Infectious Diseases Society of America (IDSA) guideline deliberately makes no recommendation for or against treating ASB in the first post-transplant month because of insufficient evidence [1]. From the second month onward, it explicitly advises against therapy in asymptomatic patients [1]. Likewise, the American Society of Transplantation—Infectious Disease (AST-ID) Community of Practice finds no demonstrable benefit beyond two months and discourages both routine screening and treatment [3]. However, the treatment of asymptomatic bacteriuria has not been shown to reduce the incidence of symptomatic urinary tract infection and may worsen microbiological outcomes over time [2].

Therefore, there is a need to further investigate and clarify the management of asymptomatic bacteriuria and candiduria in the first two months after kidney transplantation, as clinical equipoise persists: while overtreatment promotes resistance, missed opportunities may allow progression to pyelonephritis, urosepsis, or graft dysfunction [6,7,8,9].

It is necessary to emphasize that our study has significant limitations, and, in particular, it is a single-center, retrospective study with a small sample size.

In this backdrop, the present study compares outcomes in treated versus untreated kidney-transplant recipients with ASB or AC during the first two post-transplant months.

## 2. Results

We enrolled 59 patients who experienced a total of 147 episodes of asymptomatic bacteriuria or candiduria (137 and 10 episodes, respectively). Of the 59 enrolled patients, the median age was 47 years (IQR 39–54), and 49.0% were female (29/59). A total of 35 patients (59.0%) had no comorbidities. The most frequent conditions among the remaining patients were hypertension (6/59, 10.0%), type 2 diabetes mellitus (5/59, 8.5%), dyslipidemia (5/59, 8.5%) and hypothyroidism (4/59, 6.8%). Only 2 patients (3.4%) had benign prostatic hyperplasia, 1 (1.7%) had chronic HBV infection, and 1 (1.7%) an anxiety–depressive disorder. All patients received induction therapy with basiliximab plus methylprednisolone, and the majority (78%) were on maintenance immunosuppression with tacrolimus, mycophenolate and steroids.

Microbiological characteristics of isolates recovered during episodes of asymptomatic bacteriuria and candiduria episodes are summarized in Table 1.

*Escherichia coli* was the most frequently isolated organism—41% of isolates in the untreated cohort versus 38% in the treated cohort—followed by *Enterococcus faecalis*, *Klebsiella pneumoniae*, and *Enterobacter cloacae*. The predominant *Candida* species was *C. albicans*, while less frequent were *C. tropicalis*, and *C. glabrata*. Pathogen distribution did not differ significantly between the two groups (*p* = 0.7). No significant differences were observed in terms of resistance profiles (*p* = 0.9).

A univariate analysis was first carried out to compare the untreated and treated cohorts with respect to the primary endpoint—30-day mortality—and the following secondary endpoints: 90-, 180- and 360-day mortality; sepsis; septic shock; bacteremia/candidemia; hospitalization; graft loss; decline in renal function; any urinary tract infection (UTI) during the first post-transplant year; recurrent UTIs and UTIs caused by multidrug-resistant (MDR) organisms during the first post-transplant year (Table 2). No significant between-group differences were detected for any of these outcomes (Table 2).

From the multivariate analysis, no differences were observed between the two groups, nor was there a significant impact on the primary endpoint or on the secondary endpoints in terms of comorbidities (such as type 2 diabetes mellitus, obesity, arterial hypertension, dyslipidemia), of the presence of urinary devices (catheters or stents) and of the type of maintenance immunosuppressive therapy.

Mortality remained low at all time points (30, 90, 180 and 360 days). During the one-year follow-up period after the transplant, four deaths were observed in the untreated group and three deaths in the treated group. All occurred within the first 90 days after the transplant. In particular, three deaths were attributable to bacterial infections (two in the untreated group and one in the treated group), whereas the remaining deaths were attributable to one case of invasive aspergillosis, one refractory CMV infection, and the remaining two to post-transplant surgical complications.

The Kaplan–Meier curve illustrates cumulative survival in the two groups (Figure 1). Survival in both cohorts stayed near 100% throughout the first 30 days, declining only slightly toward the end; the log-rank test showed no significant difference (*p* = 0.62).

The Kaplan–Meier curve illustrates 90-day survival (Figure 1). The two trajectories overlap almost entirely, indicating that withholding treatment for asymptomatic bacteriuria or candiduria did not increase 90-day mortality (*p* = 0.61). Overall, both cohorts displayed parallel survival patterns, with no statistically significant differences across the observation period (log-rank *p* = 0.62 and *p* = 0.61, respectively).

A multivariate analysis did not identify any significant predictors of 30-day mortality. Specifically, treatment status, age, sex, and infection with multidrug-resistant (MDR) organisms were all non-significant (Table 2).

In the multivariate model for 90-day mortality, age was the only factor independently associated with death (OR 1.089; 95% CI 1.01–1.18; *p* = 0.04).

Because no deaths were recorded at 180 or 360 days, multivariate analysis was not performed for those time points.

Similarly, multivariable analysis showed that none of the covariates—antimicrobial treatment, age, sex, or isolation of multidrug-resistant pathogens—was significantly associated with the risk of sepsis.

Isolation of multidrug-resistant organisms emerged as the only independent predictor of septic shock (OR 4.639; 95% CI 1.06–20.25; *p* = 0.04), whereas antimicrobial treatment, sex, and age showed no significant effect on this outcome.

For the outcome of bacteremia or candidemia secondary to urinary tract infection, isolation of a multidrug-resistant organism was the only independent predictor (OR 3.734; 95% CI 1.45–9.61; *p* = 0.01). Antimicrobial treatment of asymptomatic urinary infections did not confer any protective effect against systemic dissemination.

In the multivariate model for hospitalization, isolation of a multidrug-resistant organism was the sole independent predictor (OR 2.183; 95% CI 1.07–4.46; *p* = 0.03). Antimicrobial treatment, sex, and age showed no significant association with this outcome.

None of the variables included in the multivariate analysis emerged as independent risk factors for developing urinary tract infections during the first post-transplant year.

In multivariate analysis, two comorbidities were independently associated with recurrent urinary tract infection: type 2 diabetes mellitus (OR 1.90; 95% CI 1.120–3.200; *p* = 0.017) and dyslipidemia (OR 11.79; 95% CI 2.010–69.120; *p* = 0.006).

Multivariate analysis revealed no significant associations between any of the evaluated variables and graft loss. Specifically, antibiotic therapy had no measurable impact on allograft survival.

Isolation of multidrug-resistant organisms was the only independent predictor of worsening renal function (OR 3.93; 95% CI 1.12–13.80; *p* = 0.03). Treatment status, sex, and age showed no association with this outcome. Accordingly, antimicrobial resistance again emerged as a negative prognostic factor.

Antimicrobial-resistance profiles of isolates detected during urinary tract infection in the first post-transplant year are presented in Table 3; multidrug-resistant strains were more prevalent in the treated group.

## 3. Discussion

The present investigation set out to clarify whether treating asymptomatic bacteriuria or candiduria during the delicate, early post-transplant window (the first two months after kidney graft implantation) alters the clinical trajectory of the recipients. When we compared those who received antimicrobial therapy with those who did not, the two cohorts proved strikingly similar across every key endpoint.

Thirty-day survival—our primary outcome—was identical in both groups, reinforcing randomized data that question the wisdom of “treating the numbers” when bacteria are cultured from an otherwise silent urinary tract [10].

Extending follow-up to three months did not change the picture. Age remained the only variable that independently influenced survival at this time point, underscoring that chronological age—with its attendant frailty and comorbidities—remains a formidable prognostic determinant in transplantation.

Although the sample size is modest, our finding that leaving asymptomatic bacteriuria (or candiduria) untreated does not imperil medium- or long-term survival supports the conclusions of several observational studies and systematic reviews [2,11,12,13].

One might argue that asymptomatic colonization could serve as a springboard to systemic infection. Yet in our cohort the decision to treat—or not treat—had no impact on the risk of subsequent developing of sepsis or septic shock, and risk of bloodstream infections. What did matter was which organism had taken hold: isolation of a multidrug-resistant (MDR) strain multiplied the odds of septic shock more than four-fold (OR 4.639; *p* = 0.04). Similar signals linking MDR pathogens to more fulminant disease have been reported in kidney transplant populations [14,15]. Moreover, MDR colonization was associated with a higher incidence of bloodstream infection irrespective of therapeutic strategy [16]. The fact that colonization by MDR organisms increased the risk of septic shock but not of sepsis is probably attributable to an initial empirical therapy that was inappropriate for the MDR organisms involved and, at the same time, not timely. It should be emphasized that this consideration is only a hypothesis and that this aspect lay outside the scope of our study.

Another theoretical benefit of treating asymptomatic bacteriuria would be prevention of future symptomatic episodes. We found no such benefit: the risk of developing a UTI—or of experiencing multiple recurrences—during the first post-transplant year was unaffected by early antimicrobial use. Instead, two metabolic disorders—type 2 diabetes and dyslipidemia—emerged as the principal drivers of recurrence. For diabetes, the mechanisms are well documented: hyperglycosuria, impaired neutrophil function and autonomic bladder dysfunction favor bacterial persistence. Dyslipidemia is less well studied, yet chronic low-grade inflammation and altered membrane lipid composition may plausibly weaken mucosal defenses [17].

Graft-loss rates did not differ between groups, further undermining the argument for routine therapy. Nonetheless, we noted that recipients in whom MDR organisms were isolated experienced a sharper decline in renal function over time, suggesting that resistant pathogens may inflict subclinical damage on the allograft [18].

In addition to the hypothesis of a microbiological correlation, the worsening of renal function could also be attributable to antibiotic-related drug toxicity, which might adversely impact graft outcomes.

Perhaps the most sobering observation was the numerical excess of MDR infections in the first year after the transplant among patients who had been treated for asymptomatic bacteriuria in the first two months after transplantation (63% vs. 46%). Although this did not reach statistical significance, it echoes the well-documented ecological cost of unnecessary antibiotics, particularly in immunosuppressed hosts where selective pressure is intense [6,19,20,21,22,23]. The specter of fostering resistance—thereby compromising future therapeutic options—argues strongly for restraint.

Our study highlights that not only would the treatment of asymptomatic bacteriuria and candiduria not provide benefits, but it could also have an adverse impact in terms of colonization/infection by MDR germs, with associated worsening of kidney function and more severe infection conditions.

Our analysis is bound by its single-center, partly retrospective design and limited sample size. Even so, the consistency of neutral (or even unfavorable) effects across multiple outcomes strengthens the message. Larger, prospective trials are needed to delineate whether any subgroup—perhaps those colonized by highly virulent MDR strains—might still benefit from targeted therapy.

## 4. Materials and Methods

We conducted a single-center, retrospective observational study at the University Hospital “Federico II” (Naples, Italy). All adult recipients who underwent kidney transplantation between 1 January 2019 and 31 July 2024 completed ≥ 12 months of follow-up, and experienced at least 1 episode of asymptomatic bacteriuria or candiduria within the first 60 days after transplantation were enrolled.

Recipients of re-transplant or pediatric patients have been excluded.

Asymptomatic bacteriuria or candiduria was defined as the isolation of ≥10^5^ CFU mL^−1^ of a pathogenic microorganism from a properly collected urine sample in the absence of any local or systemic signs or symptoms of a urinary tract infection (UTI). The diagnostic criteria for asymptomatic bacteriuria followed the most recent guidelines for immunocompromised hosts [1], whereas candiduria was defined according to international candidiasis guidelines [8]. Multidrug-resistant (MDR) organisms were defined as those non-susceptible to ≥3 antimicrobial classes [9].

Patients were stratified according to the management of the index episode—no antimicrobial therapy versus antimicrobial therapy—and the decision to treat or not was left to the discretion of the treating physician. The decision to treat or not to treat was random and not based on clinical criteria and/or risk factors. The decision to treat or not treat was made randomly at the physician’s discretion.

The primary endpoint was all-cause mortality within 30 days. Secondary endpoints comprised all-cause mortality at 90, 180 and 360 days; the incidence of sepsis or septic shock, bacteremia or candidemia and hospitalization; graft loss; decline in renal function (the worsening of renal function was defined as an increase in serum creatinine of at least 0.3 mg/dl compared to baseline) and, within the first post-transplant year, the incidence of symptomatic urinary tract infection (UTI), recurrent UTI, and UTI caused by multidrug-resistant (MDR) organisms.

The definition of sepsis and septic shock was based on what was stated by the Third International Consensus in 2016 [24].

Recurrent UTIs were defined as the presence of at least 3 UTI episodes in a year or 2 episodes within six months [1].

Categorical variables are presented as counts and percentages and were compared with the Chi-square test or Fisher’s exact test, as appropriate based on expected cell counts. Continuous variables are reported as median (inter-quartile range, IQR) and compared with the Wilcoxon rank-sum test (Mann–Whitney U test). Survival curves were generated using the Kaplan–Meier method and compared with the log-rank test. For this analysis only, in patients with multiple episodes, only the first episode was considered, and group assignment was based on the treatment status at that episode. To evaluate the marginal effect of treatment on primary and secondary endpoints, generalized estimating equations (GEE) with a logit link and exchangeable correlation structure were employed, accounting for correlation between repeated observations within the same patient and providing population-averaged odds ratio estimates. In the first phase, for each endpoint, a univariate analysis including only the treatment variable was performed to assess its crude association with the outcome. Subsequently, for all endpoints, a multivariable model was fitted including treatment and a set of covariates (age, sex, MDR/non-MDR status) selected for their clinical plausibility, prior evidence and potential role as confounders of the treatment–outcome association. These covariates were identified from a full model including all clinically relevant variables for the primary endpoint. The treatment variable was included a priori in all models as the main exposure of interest. The same set of covariates was applied to all primary and secondary endpoints to ensure consistency of adjustment, comparability of effect estimates and to avoid outcome-specific model selection. A two-sided *p*-value < 0.05 was considered statistically significant for all analyses.

The software used was R version 4.4.3, R Foundation for Statistical Computing Vienna, Austria, https://www.R-project.org.

The study was conducted in compliance with the Declaration of Helsinki and the principles of good clinical practice.

## 5. Conclusions

Taken together, our data suggest that withholding antibiotics for asymptomatic bacteriuria or candiduria in the early post-transplant period is safe and, by sparing unnecessary drug exposure, may help curb the emergence of multidrug-resistant organisms. Mortality through one year, as well as rate of sepsis, septic shock, bloodstream infection, hospitalization, UTI incidence and recurrence, decline in renal function and graft loss were all unaffected by treatment status. In contrast, the presence of MDR colonization itself portended worse infectious outcomes, underscoring the need for close surveillance rather than blanket therapy. Until robust evidence to the contrary emerges, a watch-and-wait approach—tempered by antimicrobial stewardship principles and attentive follow-up of high-risk metabolic or MDR-colonized recipients—appears the most judicious course.

We are aware of the limitations of our study, particularly the single-center retrospective nature and the small sample size; therefore, further in-depth studies are needed.

## Figures and Tables

**Figure 1 antibiotics-14-01155-f001:**
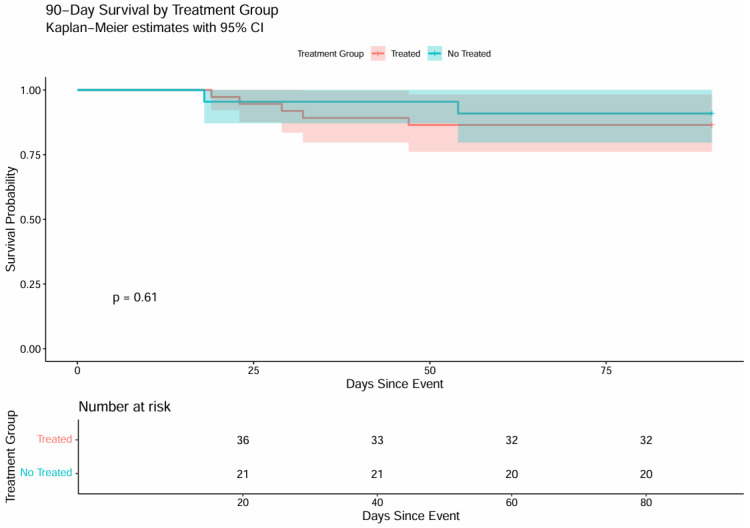
90-day survival by treatment group—Kaplan–Meier.

**Table 1 antibiotics-14-01155-t001:** Characteristics of asymptomatic bacteriuria/candiduria.

		Untreated Episodes *n* = 95	Treated Episodes *n* = 52	*p*-Value
Asymptomatic bacteriuria		90 (94.7%)	47 (90.4%)	0.3
Asymptomatic candiduria		5 (5.3%)	5 (9.6%)	0.3
Microbiological agents isolated ^1^, *n* (%)				0.7
	Resistance profile			0.9
*Candida albicans*		2 (2.1%)	4 (7.7%)	
*Candida glabrata*		0 (0.0%)	1 (1.9%)	
*Candida tropicalis*		3 (3.2%)	0 (0.0%)	
*Citrobacter braakii*		1 (1.1%)	0 (0.0%)	
*Citrobacter braakii*	ESBL	2 (2.1%)	0 (0.0%)	
*Citrobacter farmeri*	ESBL	1 (1.1%)	0 (0.0%)	
*Citrobacter* spp.		0 (0.0%)	1 (1.9%)	
*Enterobacter cloacae*		2 (2.1%)	0 (0.0%)	
*Enterobacter cloacae*	ESBL	1 (1.1%)	1 (1.9%)	
*Enterobacter cloacae*	CR	1 (1.1%)	0 (0.0%)	
*Enterococcus faecalis*		17 (17.9%)	11 (21.2%)	
*Enterococcus faecalis*	ampi-R	2 (2.1%)	1 (1.9%)	
*Enterococcus faecium*	ampi-R	3 (3.2%)	2 (3.8%)	
*Enterococcus faecium*	VRE	0 (0.0%)	1 (1.9%)	
*Escherichia coli*		23 (24.2%)	10 (19.2%)	
*Escherichia coli*	ESBL	16 (16.8%)	10 (19.2%)	
*Klebsiella pneumoniae*		11 (11.6%)	5 (9.6%)	
*Klebsiella pneumoniae*	ESBL	2 (2.1%)	2 (3.8%)	
*Morganella morganii*	ESBL	1 (1.1%)	1 (1.9%)	
*Proteus mirabilis*	ESBL	1 (1.1%)	0 (0.0%)	
*Providencia rettgeri*		1 (1.1%)	0 (0.0%)	
*Pseudomonas aeruginosa*		1 (1.1%)	0 (0.0%)	
*Pseudomonas aeruginosa*	MDR	2 (2.1%)	2 (3.8%)	
*Stenotrophomonas maltophilia*		1 (1.1%)	0 (0.0%)	
*Streptococcus gallolyticus*		1 (1.1%)	0 (0.0%)	
Any MDR pathogen		13 (13.7%)	8 (15.4%)	

^1^ Isolated from urine culture. ESBL: extended-spectrum beta-lactamases; CR: carbapenem-resistant; ampi-R: ampicillin-resistant; VRE: vancomycin-resistant *Enterococci*; MDR: multidrug-resistant.

**Table 2 antibiotics-14-01155-t002:** Univariate and multivariate analysis: asymptomatic bacteriuria/candiduria untreated vs. treated.

	Untreated Episodes *n* = 95	Treated Episodes *n* = 52	Univariate Analysis OR [95% CI] *p*-Value	Multivariate Analysis OR [95% CI] *p*-Value			
				Treatment	Age	Sex	Infection with Multidrug Resistant (MDR)
30-day mortality	2 (2.1%)	2 (3.84%)	1.176 [0.688–2.010] 0.55	0.825 [0.19–3.63] 0.80	1.183 [0.99–1.42] 0.07	2.488 [0.31–19.90] 0.39	0.542 [0.16–1.88] 0.33
90-day mortality	2 (2.1%)	1 (1.92%)	0.912 [0.627–1.327] 0.63	0.747 [0.42–1.33] 0.32	1.089 [1.01–1.18] 0.04	2.612 [0.39–17.68] 0.32	0.586 [0.22–1.57] 0.29
Hospitalization	28 (29%)	21 (40%)	1.640 [0.861–3.124] 0.13	1.499 [0.77–2.92] 0.23	1.309 [0.68–2.51] 0.42	1.017 [0.99–1.05] 0.28	2.183 [1.07–4.46] 0.03
Bacteremia/ Candidemia	14 (15%)	5 (9.6%)	0.646 [0.205–2.039] 0.46	0.606 [0.20–1.85] 0.38	1.022 [0.97–1.07] 0.41	1.523 [0.61–3.84] 0.37	3.734 [1.45–9.61] 0.01
Sepsis	10 (11%)	6 (12%)	1.155 [0.358–3.726] 0.81	1.114 [0.35–3.53] 0.85	1.684 [0.69–4.13] 0.25	1.008 [0.97–1.05] 0.73	1.063 [0.40–2.83] 0.90
Septic shock	3 (3.2%)	4 (7.7%)	2.550 [0.550–11.860] 0.23	2.377 [0.50–11.28] 0.28	1.017 [0.95–1.10] 0.65	1.177 [0.31–4.44] 0.81	4.639 [1.06–20.25] 0.04
Graft loss	7 (7.4%)	6 (12%)	1.353 [0.563–3.253] 0.5	1.360 [0.54–3.45] 0.52	1.039 [0.97–1.11] 0.28	2.048 [0.45–9.33] 0.35	0.619 [0.13–2.88] 0.54
Renal function worsening	11 (12%)	5 (9.6%)	0.819 [0.285–2.350] 0.71	0.675 [0.25–1.83] 0.44	1.061 [1.00–1.12] 0.04	1.704 [0.48–6.07] 0.41	3.930 [1.12–13.80] 0.03
Infection in first year	57 (60%)	30 (59%)	0.988 [0.967–1.010] 0.29	0.998 [0.99–1.00] 0.19	0.979 [0.94–1.03] 0.37	1.080 [0.37–3.15] 0.89	0.996 [0.99–1.00] 0.20
Recurrent UTI	16 (17%)	12 (23%)	1.474 [0.205–2.039] 0.38	1.234 [0.59–2.60] 0.58	1.037 [0.99–1.08] 0.09	1.205 [0.66–2.20] 0.54	0.763 [0.35–1.67] 0.50
MDR risk	44 (46%)	33 (63%)	0.995 [0.98–1.01] 0.072	0.978 [0.93–1.02] 0.33	1.032 [0.37–2.90] 0.95	1.023 [0.98–1.07] 0.31	1.040 [0.97–1.12] 0.30

OR: odds ratio (untreated vs. treated); CI: confidence interval; UTI: urinary tract infection; MDR: multidrug-resistant.

**Table 3 antibiotics-14-01155-t003:** Characteristics and resistance profile of microorganisms isolated in the follow-up period.

		Untreated Asymptomatic Bacteriuria/Candiduria *n* = 44 (46%)	Treated Asymptomatic Bacteriuria/Candiduria *n* = 33 (63%)	*p*-Value
Microbiological agents isolated ^1^, *n* (%)				0.072
	Resistance profile			
*Candida albicans*	FLU-R	0 (0.0%)	3 (9.1%)	
*Citrobacter* spp.	ESBL	3 (6.8%)	0 (0.0%)	
*Enterococcus faecalis*	ampi-R	6 (13.6%)	2 (6.1%)	
*Enterococcus faecalis*	VRE	2 (4.5%)	2 (6.1%)	
*Enterococcus faecium*	VRE	3 (6.8%)	2 (6.1%)	
*Escherichia coli*	ESBL	9 (20.5%)	5 (15.2%)	
*Escherichia coli*	CR	3 (6.8%)	2 (6.1%)	
*Escherichia coli*	NDM	1 (2.3%)	0 (0.0%)	
*Klebsiella pneumoniae*	ESBL	12 (27.3%)	3 (9.1%)	
*Klebsiella pneumoniae*	CR	5 (11.4%)	7 (21.2%)	
*Klebsiella pneumoniae*	NDM	0 (0.0%)	1 (3.0%)	
*Pseudomonas aeruginosa*	DTR	0 (0.0%)	6 (18.2%)	

^1^ Isolated from urine culture. FLU-R: fluconazole-resistant; ESBL: extended-spectrum beta-lactamases; ampi-R: ampicillin-resistant; VRE: vancomycin-resistant *Enterococci*; CR: carbapenem-resistant; NDM: New Delhi Metallo beta-lactamase; DTR: Difficult-to-Treat Resistance.

## Data Availability

The data are available upon request from the corresponding author.

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
