# Peer review of "Clinical Impact of Treating Versus Not Treating Asymptomatic Bacteriuria/Candiduria in the First Two Months After Kidney Transplantation"

_antibiotics, 2025, doi:10.3390/antibiotics14111155_

Round 1
Reviewer 1 Report
Comments and Suggestions for Authors
Dear Editor, Dear authors,
this is an interesting retrospective study regarding a topic of debate in the transplant and ID community. In the manuscript the introduction and methods sections are adequate and accurately describe what was the intention and what was done.
There is one major comment in regards to the study design - and that is the description of the studied population - no major patient characteristics, other than age and sex, were described. Including these variables is vital for a better understanding of the population, also to try and understand why some people were treated and others were not but also to display whether something in their inherent characteristics put them at risk of sepsis/bacteremia, graft function worsening etc. Additionally, decline in renal function was not defined.
Another useful variable would be whether the assigned treatment in the treated group was adequate according to the microbial isolate.
English language and the writing are of high quality and merit no changes.
Minor Comment
Table 2 - results significant for hte hypothesis should be visually emphasized (both positive and negative findings)
Author Response
Responses to Reviewer 1:
Dear Reviewer, first of all, I sincerely thank you for your reviews and, above all, for your advice and suggestions. With your essential and fundamental contribution, the manuscript has taken on a whole new light. Below, I am sending you the response to all the requested revisions.
“This is an interesting retrospective study regarding a topic of debate in the transplant and ID community. In the manuscript the introduction and methods sections are adequate and accurately describe what was the intention and what was done.
There is one major comment in regards to the study design - and that is the description of the studied population - no major patient characteristics, other than age and sex, were described. Including these variables is vital for a better understanding of the population, also to try and understand why some people were treated and others were not but also to display whether something in their inherent characteristics put them at risk of sepsis/bacteremia, graft function worsening etc.
I have included in the text the role of comorbidities (such as type 2 diabetes mellitus, obesity, arterial hypertension, and dyslipidemia), of the presence of urinary devices (catheters and stents) and the role of the type of immunosuppressive therapy in relation to the two groups and their possible impact on the primary endpoint and secondary endpoints. In particular, we had already carried out the multivariate analysis; however, due to the limited number of tables and figures, we had not included the results in the tables. For this reason, we have included in the text the following sentence: “In the multivariate analysis, no differences were observed between the two groups, nor was a significant impact observed on the primary endpoint and secondary endpoints in terms of comorbidities (such as type 2 diabetes mellitus, obesity, arterial hypertension, dyslipidemia) and the type of maintenance immunosuppressive therapy”, as highlighted in the text.
Additionally, decline in renal function was not defined.
I have defined the worsening of kidney function as an increase in serum creatinine of at least 0.3 mg/dl compared to baseline, as highlighted in the text.
Another useful variable would be whether the assigned treatment in the treated group was adequate according to the microbial isolate.
In this regard, the patients who were part of the treated group all underwent targeted and optimal therapy for the isolated pathogen identified.
English language and the writing are of high quality and merit no changes.
Minor Comment
Table 2 - results significant for hte hypothesis should be visually emphasized (both positive and negative findings)”.
I have highlighted all the results.
I thank you for your essential contribution and fundamental support.
Awaiting your response, I thank you for your courtesy, cooperation, and availability.
Reviewer 2 Report
Comments and Suggestions for Authors
The present manuscript aimed to evaluate whether treating, versus not treating, ASB and ASC episodes in the first two months after kidney transplantation influences clinical outcomes and the emergence of multidrug-resistant (MDR) infections. This paper is of high public health relevance; however, there are many areas of concern that need to be addressed to improve the quality of this manuscript.
Areas of concern:
Abstract
- Please, methods and results are consistently described in past tense
- Please, consider avoiding subjective or speculative language
- Please, consider including the total number of patients (n) and episodes (n) early in the Results to provide study scope immediately
- Please, consider including precise time frames and statistical details succinctly
- Introduction
- Lines 42-47: Please, consider briefly emphasizing the clinical and antimicrobial stewardship implications in the opening paragraph to integrate the study’s importance for Antibiotics
- Lines 64-65: Please, consider explicitly stating the research gap of your study
- Lines 62-63: Please, consider adding a bridging phrase before the final sentence to improve transition to your study objective
- Please, consider verifying that reference formatting aligns with MDPI’s Vancouver numerical citation style
- Results
- Please, consider combining sentences related to deaths and causes summarized together
- Results should be factual and objective, therefore, consider moving interpretive comments about clinical meaning or policy implications to the Discussion section
- Please, consider organizing results into subheadings for clarity and logical flow
- Discussion
- Please, consider clarifying your study’s novelty and contribution while discussing your findings (for e.g., early post-transplant timing, inclusion of both bacteriuria and candiduria, MDR impact, or analysis of metabolic comorbidities)
- Please, consider highlighting more explicitly as a distinctive outcome of your study, the link between MDR colonization and septic shock given its clinical relevance
- Please, consider condensing the section on metabolic comorbidities to improve flow
- Materials and Methods
- The term “prospective–retrospective observational study” is ambiguous, please clarify whether it was a “mixed prospective and retrospective cohort study” or a “hybrid cohort design”
- It is good to specify the time frame separation: which portion of data was retrospective (2019–2023?) and which was prospective (2023–2024?).
- Please, consider clarifying whether re-transplant or pediatric recipients were excluded, and how recurrent episodes were handled
- Please, consider defining how “sepsis” and “UTI recurrence” were operationalized
- Please, consider specifying the software (name, version, and manufacturer) during statistical analysis
- Please, consider specifying the ethics committee name, approval number, and informed consent procedure
- Conclusions
- Please, consider adding a sentence about limitations or future research at the end of your conclusion.
Author Response
Responses to Reviewer 2:
Dear Reviewer, first of all, I sincerely thank you for your reviews and, above all, for your advice and suggestions. With your essential and fundamental contribution, the manuscript has taken on a whole new light. Below, I am sending you the response to all the requested revisions.
The present manuscript aimed to evaluate whether treating, versus not treating, ASB and ASC episodes in the first two months after kidney transplantation influences clinical outcomes and the emergence of multidrug-resistant (MDR) infections. This paper is of high public health relevance; however, there are many areas of concern that need to be addressed to improve the quality of this manuscript.
Areas of concern:
Abstract
Please, methods and results are consistently described in past tense
Please, consider avoiding subjective or speculative language
Please, consider including the total number of patients (n) and episodes (n) early in the Results to provide study scope immediately
Please, consider including precise time frames and statistical details succinctly
I have carried out the indicated revisions.
Introduction
Lines 42-47: Please, consider briefly emphasizing the clinical and antimicrobial stewardship implications in the opening paragraph to integrate the study’s importance for Antibiotics
Lines 64-65: Please, consider explicitly stating the research gap of your study
Lines 62-63: Please, consider adding a bridging phrase before the final sentence to improve transition to your study objective
Please, consider verifying that reference formatting aligns with MDPI’s Vancouver numerical citation style
I have carried out the indicated revisions.
Results
Please, consider combining sentences related to deaths and causes summarized together
Results should be factual and objective, therefore, consider moving interpretive comments about clinical meaning or policy implications to the Discussion section
Please, consider organizing results into subheadings for clarity and logical flow
I have carried out the indicated revisions.
Discussion
Please, consider clarifying your study’s novelty and contribution while discussing your findings (for e.g., early post-transplant timing, inclusion of both bacteriuria and candiduria, MDR impact, or analysis of metabolic comorbidities)
Please, consider highlighting more explicitly as a distinctive outcome of your study, the link between MDR colonization and septic shock given its clinical relevance
Please, consider condensing the section on metabolic comorbidities to improve flow
I have carried out the indicated revisions.
Materials and Methods
The term “prospective–retrospective observational study” is ambiguous, please clarify whether it was a “mixed prospective and retrospective cohort study” or a “hybrid cohort design”
It is good to specify the time frame separation: which portion of data was retrospective (2019–2023?) and which was prospective (2023–2024?).
Please, consider clarifying whether re-transplant or pediatric recipients were excluded, and how recurrent episodes were handled
Please, consider defining how “sepsis” and “UTI recurrence” were operationalized
Please, consider specifying the software (name, version, and manufacturer) during statistical analysis
Please, consider specifying the ethics committee name, approval number, and informed consent procedure
I have carried out the indicated revisions.
Conclusions
Please, consider adding a sentence about limitations or future research at the end of your conclusion.
I have carried out the indicated revisions.
I thank you for your essential contribution and fundamental support.
Awaiting your response, I thank you for your courtesy, cooperation, and availability.
Reviewer 3 Report
Comments and Suggestions for Authors
A useful single-center study (59 patients; 147 episodes) addressing an important clinical question, whether treating early asymptomatic bacteriuria/candiduria in kidney transplant recipients improves outcomes. The data suggest no clinical benefit from routine treatment and a possible signal toward more MDR emergence in treated episodes. The study is well done overall but needs clarification on several methodological points and some tempering of causal language.
Major comments
- The decision to treat was at clinicians’ discretion. This raises substantial confounding risk (sicker or higher-risk patients may be more likely to be treated). Please: (a) describe treatment decision criteria used at your center; (b) present baseline comparisons for key confounders (catheter status, prior antibiotics, diabetes, immunosuppression intensity, prior MDR colonization) between treated vs untreated episodes; and (c) perform adjusted analyses (multivariable models including those covariates) or a propensity score approach (matching or IPTW).
- You analyzed 147 episodes from 59 patients and mention that for survival only the first episode was used, while GEE were used elsewhere. Please explicitly state: which analyses are per-episode vs per-patient, how correlations were modelled, and whether sensitivity analyses using only first episodes were performed for all outcomes. Add a small flowchart showing patient → episode counts.
- Models include treatment, age, sex, MDR status , but other clinically important confounders (diabetes, dyslipidemia, catheter presence, prior UTI history, baseline renal function, induction/maintenance immunosuppression intensity) should be considered. Either justify omission or re-run models with those covariates where possible.
- With 59 patients and low mortality, the study is underpowered to detect modest differences in mortality. Please add 95% CIs for the main effect estimates (many ORs printed without clear CI formatting), and add a short discussion of the study’s power/precision limits. Consider reframing conclusions accordingly.
- Treated episodes had a higher proportion of subsequent MDR infections (63% vs 46%). Please explore temporality (did MDR appear after treatment, or were MDR organisms present at index?) and, if possible, report whether treated patients received broader agents that might select for resistance. Add a short discussion on plausible mechanisms and stewardship implications.
- Tables are informative but have formatting issues (commas/decimal places, alignment, some p/CI text oddities). Ensure every table shows denominators, clear column headers, and 95% CIs. Add a small supplementary table with organism-level susceptibility data (it will be useful).
Minor comments
- Methods: clarify whether candiduria threshold was the same CFU≥10^5 and cite the guideline for immunocompromised hosts used.
- Results: tighten wording where numbers are repeated (e.g., Figure 1 text lines were duplicative).
- Discussion: soften causal wording; use “associated with” rather than “led to” or “increases” unless supported by adjusted causal analyses.
- Typos: a few small grammar/formatting glitches (e.g., “reccom mend” I spotted one or two).
Author Response
Responses to Reviewer 3:
Dear Reviewer, first of all, I sincerely thank you for your reviews and, above all, for your advice and suggestions. With your essential and fundamental contribution, the manuscript has taken on a whole new light. Below, I am sending you the response to all the requested revisions.
A useful single-center study (59 patients; 147 episodes) addressing an important clinical question, whether treating early asymptomatic bacteriuria/candiduria in kidney transplant recipients improves outcomes. The data suggest no clinical benefit from routine treatment and a possible signal toward more MDR emergence in treated episodes. The study is well done overall but needs clarification on several methodological points and some tempering of causal language.
Major comments
The decision to treat was at clinicians’ discretion. This raises substantial confounding risk (sicker or higher-risk patients may be more likely to be treated). Please: (a) describe treatment decision criteria used at your center; (b) present baseline comparisons for key confounders (catheter status, prior antibiotics, diabetes, immunosuppression intensity, prior MDR colonization) between treated vs untreated episodes; and (c) perform adjusted analyses (multivariable models including those covariates) or a propensity score approach (matching or IPTW).
The decision to treat or not to treat was random and not based on clinical criteria and/or risk factors, nor on severity, since in all cases it involved asymptomatic bacteriuria. I have specified this in the text and in particular in “Materials and Methods”, as highlighted.
I have included in the text the role of comorbidities (such as type 2 diabetes mellitus, obesity, arterial hypertension, and dyslipidemia), the role of the presence of urinary devices (catheters and stents) and the role of the type of immunosuppressive therapy in relation to the two groups and their possible impact on the primary endpoint and secondary endpoints. In particular, we had already carried out the multivariate analysis; however, due to the limited number of tables and figures, we had not included the results in the tables. For this reason, we have included in the text the following sentence: “In the multivariate analysis, no differences were observed between the two groups, nor was a significant impact observed on the primary endpoint and secondary endpoints in terms of comorbidities (such as type 2 diabetes mellitus, obesity, arterial hypertension, dyslipidemia) and the type of maintenance immunosuppressive therapy”, as highlighted in the text.
You analyzed 147 episodes from 59 patients and mention that for survival only the first episode was used, while GEE were used elsewhere. Please explicitly state: which analyses are per-episode vs per-patient, how correlations were modelled, and whether sensitivity analyses using only first episodes were performed for all outcomes. Add a small flowchart showing patient → episode counts.
I have made sure to specify everything in the text.
Models include treatment, age, sex, MDR status , but other clinically important confounders (diabetes, dyslipidemia, catheter presence, prior UTI history, baseline renal function, induction/maintenance immunosuppression intensity) should be considered. Either justify omission or re-run models with those covariates where possible.
I have included in the text the role of comorbidities (such as type 2 diabetes mellitus, obesity, arterial hypertension, and dyslipidemia), the role of the presence of urinary devices (catheters and stents) and the role of the type of immunosuppressive therapy in relation to the two groups and their possible impact on the primary endpoint and secondary endpoints. In particular, we had already carried out the multivariate analysis; however, due to the limited number of tables and figures, we had not included the results in the tables. For this reason, we have included in the text the following sentence: “In the multivariate analysis, no differences were observed between the two groups, nor was a significant impact observed on the primary endpoint and secondary endpoints in terms of comorbidities (such as type 2 diabetes mellitus, obesity, arterial hypertension, dyslipidemia) and the type of maintenance immunosuppressive therapy”, as highlighted in the text.
With 59 patients and low mortality, the study is underpowered to detect modest differences in mortality. Please add 95% CIs for the main effect estimates (many ORs printed without clear CI formatting), and add a short discussion of the study’s power/precision limits. Consider reframing conclusions accordingly.
I have taken care of what was requested; in particular, the tables show the confidence intervals, and at the end of the discussion, the limitations of our study are extensively specified. Specifically, at the very end of the discussion, it is stated that our study has significant limitations due to the small sample size and the retrospective and single-center nature of the study.
Treated episodes had a higher proportion of subsequent MDR infections (63% vs 46%). Please explore temporality (did MDR appear after treatment, or were MDR organisms present at index?) and, if possible, report whether treated patients received broader agents that might select for resistance. Add a short discussion on plausible mechanisms and stewardship implications.
As already reported in the text, a follow-up period of approximately one year post-transplant was observed for each patient, so the detection of MDR germs is observed within a one-year follow-up period from kidney transplantation. The possible correlation between infection/colonization by MDR germs during the post-transplant follow-up period and the management of asymptomatic bacteriuria does not lie in the type of antibiotic therapy used, which is always targeted, but in relation to the use of antibiotic therapy in the management of asymptomatic bacteriuria. In particular, patients who underwent antibiotic therapy for asymptomatic bacteriuria had a higher risk of developing infection/colonization by MDR germs.
Tables are informative but have formatting issues (commas/decimal places, alignment, some p/CI text oddities). Ensure every table shows denominators, clear column headers, and 95% CIs. Add a small supplementary table with organism-level susceptibility data (it will be useful).
I have made the requested changes and revisions.
Minor comments
Methods: clarify whether candiduria threshold was the same CFU≥10^5 and cite the guideline for immunocompromised hosts used.
I have made the requested changes, and in particular in "Materials and Methods" the significant load of candiduria is specified, specifically: "Asymptomatic bacteriuria or candiduria was defined as the isolation of ≥10⁵ CFU mL⁻¹ of a pathogenic microorganism from a properly collected urine sample in the absence of any local or systemic signs or symptoms of a urinary-tract infection (UTI)."
Results: tighten wording where numbers are repeated (e.g., Figure 1 text lines were duplicative).
I have made the requested revisions and modifications.
Discussion: soften causal wording; use “associated with” rather than “led to” or “increases” unless supported by adjusted causal analyses.
I have made the requested changes and revisions, as highlighted in the text.
Typos: a few small grammar/formatting glitches (e.g., “reccommend” I spotted one or two).
I have made the requested changes and revisions, as highlighted in the text.
I thank you for your essential contribution and fundamental support.
Awaiting your response, I thank you for your courtesy, cooperation, and availability.
Round 2
Reviewer 1 Report
Comments and Suggestions for Authors
Dear authors,
thank you for your timely response.
To summarize, this is a paper about management of asymptomatic bacteriuria and candiduria in early kidney tx. The results suggest that treatment has no effect on the selected outcomes - mortality, graf t function worsening, sepsis, septic shock, admission, recurrent uti, graft loss. Mortality was however low, as were serious events such as septic shock. There are several important and interesting findings, particularly regarding MDR strains and their negative effects - increased hospitalization rates and graft functional decline. In this study treating the initial asymptomatic infection appears not to have affected these outcomes.
In the first round of reviews I have asked for additional information on the patients included and this has been provided in the text. Also other minor remarks have been added.
A nice new table 3 showing the resistance profile isolated in the follow up period shows emergence of some new organisms, particularly MDR strains in the follow-up period, more frequently in the treated group. This is a really important table illustrating the risks of antibiotic (over)use.
Major revisions
In the revised manuscript the authors state in the methods section, line 241: The decision to treat or not to treat was random and not based on clinical criteria and/or risk factors.
How was randomization performed? This is described to be a retrospective study, however, it would appear from this explanation that randomization was performed? Please describe in full detail the randomization process. Or was it the case that treatment was left to the discretion of the treating physician? This section needs thorough revision and exact description of the process.
An inconsistency in table 2. The authors state 4 people died within the first 30 days, 3 in the first ninety and 0 in the first 180/360 days? I understand that the authors mean from the rpevious to the current timepoint, but this is misleading, as usually 6 month mortality means from day 1 do day 360. Please rephrase the statement and remove the two rows as they are empty, do not add information and are not necessary and could help shorten an already long table.
Another possible point to address in the discussion is the relation of MDR isolates and declining graft function - could the increased use of antibiotics (particularly those for treatment of MDR) have also contributed to worsening renal function?
Author Response
Response to Reviewer 1:
First of all, I thank you for the essential and fundamental suggestions and revisions. Thanks to your contribution, the paper has taken on a whole new perspective and has improved. I have made the requested changes and revisions, as indicated in the text.
Comments and Suggestions for Authors
Dear authors,
thank you for your timely response.
To summarize, this is a paper about management of asymptomatic bacteriuria and candiduria in early kidney tx. The results suggest that treatment has no effect on the selected outcomes - mortality, graf t function worsening, sepsis, septic shock, admission, recurrent uti, graft loss. Mortality was however low, as were serious events such as septic shock. There are several important and interesting findings, particularly regarding MDR strains and their negative effects - increased hospitalization rates and graft functional decline. In this study treating the initial asymptomatic infection appears not to have affected these outcomes.
In the first round of reviews I have asked for additional information on the patients included and this has been provided in the text. Also other minor remarks have been added.
A nice new table 3 showing the resistance profile isolated in the follow up period shows emergence of some new organisms, particularly MDR strains in the follow-up period, more frequently in the treated group. This is a really important table illustrating the risks of antibiotic (over)use.
Major revisions
In the revised manuscript the authors state in the methods section, line 241: The decision to treat or not to treat was random and not based on clinical criteria and/or risk factors.
How was randomization performed? This is described to be a retrospective study, however, it would appear from this explanation that randomization was performed? Please describe in full detail the randomization process. Or was it the case that treatment was left to the discretion of the treating physician? This section needs thorough revision and exact description of the process.
The decision to treat or not treat was made randomly at the physician's discretion. I have included this in the text, as you requested.
An inconsistency in table 2. The authors state 4 people died within the first 30 days, 3 in the first ninety and 0 in the first 180/360 days? I understand that the authors mean from the rpevious to the current timepoint, but this is misleading, as usually 6 month mortality means from day 1 do day 360. Please rephrase the statement and remove the two rows as they are empty, do not add information and are not necessary and could help shorten an already long table.
I have made the changes to the text, replacing it with the following sentence “…During the one-year follow-up period after the transplant, 4 deaths were observed in the untreated group and 3 deaths in the treated group. All occurred within the first 90 days after the transplant….”.
As reported in the manuscript, I have made the changes to Table 2 as indicated and as mentioned in the text.
Another possible point to address in the discussion is the relation of MDR isolates and declining graft function - could the increased use of antibiotics (particularly those for treatment of MDR) have also contributed to worsening renal function?
I have taken care to add the indicated topic to the discussion, as stated in the manuscript.
Thank you for everything!
Your contribution has been essential and fundamental!
Thank you very much!
Reviewer 2 Report
Comments and Suggestions for Authors
Dear Authors,
It was a pleasure to review your manuscript entitled “Clinical Impact of Treating versus Not Treating Asymptomatic Bacteriuria/Candiduria in the First Two Months after Kidney Transplantation.” The topic is of great public health relevance, and your work offers valuable insights.
The reviewers have provided constructive feedback with the aim of further enhancing the quality of your manuscript. I encourage you to carefully consider their comments and respond thoughtfully. When addressing the reviewer’s suggestions, please:
- Highlight any changes made to the manuscript for ease of review.
- If there is disagreement with a comment, kindly explain your rationale or provide a suitable rebuttal where appropriate.
I look forward to seeing your revised submission.
Best regards,
Areas of concern:
- Introduction
- Lines 62-63: Please, consider adding a bridging phrase before the final sentence to improve transition to your study objective
- Lines 64-65: Please, consider explicitly stating the research gap of your study
- Results
- Please, consider organizing results into subheadings for clarity and logical flow
- Discussion
- Please, consider clarifying your study’s novelty and contribution while discussing your findings (for e.g., early post-transplant timing, inclusion of both bacteriuria and candiduria, MDR impact, or analysis of metabolic comorbidities)
- Please, consider highlighting more explicitly as a distinctive outcome of your study, the link between MDR colonization and septic shock given its clinical relevance
- Please, consider condensing the section on metabolic comorbidities to improve flow
- Materials and Methods
- Please, sub-divide into subheadings for better flow and clarity
- The term “prospective–retrospective observational study” is ambiguous, please clarify whether it was a “mixed prospective and retrospective cohort study” or a “hybrid cohort design”
- It is good to specify the time frame separation: which portion of data was retrospective (2019–2023?) and which was prospective (2023–2024?).
- Please, consider clarifying whether re-transplant or pediatric recipients were excluded, and how recurrent episodes were handled
- Please, consider defining how “sepsis” and “UTI recurrence” were operationalized
- Please, consider specifying the software (name, version, and manufacturer) during statistical analysis
- Conclusions
- Please, consider adding a sentence about limitations or future research at the end of your conclusion.
Author Response
Response to Reviewer 2 (Round 2):
First of all, I thank you for the essential and fundamental suggestions and revisions. Thanks to your contribution, the paper has taken on a whole new perspective and has improved. I have made the requested changes and revisions, as indicated in the text.
Dear Authors,
It was a pleasure to review your manuscript entitled “Clinical Impact of Treating versus Not Treating Asymptomatic Bacteriuria/Candiduria in the First Two Months after Kidney Transplantation.” The topic is of great public health relevance, and your work offers valuable insights.
The reviewers have provided constructive feedback with the aim of further enhancing the quality of your manuscript. I encourage you to carefully consider their comments and respond thoughtfully. When addressing the reviewer’s suggestions, please:
Highlight any changes made to the manuscript for ease of review.
If there is disagreement with a comment, kindly explain your rationale or provide a suitable rebuttal where appropriate.
I look forward to seeing your revised submission.
Best regards,
Areas of concern:
Introduction
Lines 62-63: Please, consider adding a bridging phrase before the final sentence to improve transition to your study objective
I have made the requested revisions, as highlighted in the text: “…Therefore, there is a need to further investigate and clarify the management of asymptomatic bacteriuria and candiduria in the first two months after kidney transplan-tation, as cClinical equipoise persists: while overtreatment promotes resistance, missed opportunities may allow progression to pyelonephritis, urosepsis, or graft dysfunction [6-9]….”.
Lines 64-65: Please, consider explicitly stating the research gap of your study
I have carried out the requested revisions, as indicated in the text: “…It is necessary to emphasize that our study has significant limitations, and in particular, it is a single-center, retrospective study with a small sample size…”.
Results
Please, consider organizing results into subheadings for clarity and logical flow
The results have already been fairly summarized both in the text and in the table.
Discussion
Please, consider clarifying your study’s novelty and contribution while discussing your findings (for e.g., early post-transplant timing, inclusion of both bacteriuria and candiduria, MDR impact, or analysis of metabolic comorbidities)
I have made the requested revisions, as indicated in the text.
Please, consider highlighting more explicitly as a distinctive outcome of your study, the link between MDR colonization and septic shock given its clinical relevance.
I have made the requested revisions, as indicated in the text.
Please, consider condensing the section on metabolic comorbidities to improve flow.
I have made the requested revisions, as indicated in the text.
Materials and Methods
Please, sub-divide into subheadings for better flow and clarity
The term “prospective–retrospective observational study” is ambiguous, please clarify whether it was a “mixed prospective and retrospective cohort study” or a “hybrid cohort design”
It is good to specify the time frame separation: which portion of data was retrospective (2019–2023?) and which was prospective (2023–2024?).
I have clarified that it is a retrospective study, making the revisions and corrections.
Please, consider clarifying whether re-transplant or pediatric recipients were excluded, and how recurrent episodes were handled.
I have carried out the requested revision, specifying and inserting into the text "Recipients of re-transplant or pediatric patients have been excluded".
Recurring episodes were considered as separate episodes.
Please, consider defining how “sepsis” and “UTI recurrence” were operationalized
I have proceeded to integrate what you suggested, as reported in the text: “The definition of sepsis and septic shock was based on what was stated by the Third International Consensus in 2016…Recurrent UTIs were defined as the presence of at least 3 UTI episodes in a year or 2 episodes within six months…”
Please, consider specifying the software (name, version, and manufacturer) during statistical analysis
I have taken care of what was requested; in particular, I have included in the text the type of software: "R version 4.4.3, R Foundation for Statistical Computing Vienna, Austria, URL: https://www.R-project.org."
Conclusions
Please, consider adding a sentence about limitations or future research at the end of your conclusion.
I have carried out the requested revisions, as stated in the text: "…We are aware of the limitations of our study, particularly its single-center retrospective nature and the small sample size; therefore, further in-depth studies are needed…"
Thank you for everything!
Your contribution has been essential and fundamental!
Thank you very much!
Reviewer 3 Report
Comments and Suggestions for Authors
Accept in present form
Author Response
Response to Reviewer 3 (Round 3):
I thank you for the essential and fundamental suggestions and revisions. Thanks to your contribution, the paper has taken on a whole new perspective and has improved.
Thank you for everything!
Your contribution has been essential and fundamental!
Thank you very much!
Round 3
Reviewer 1 Report
Comments and Suggestions for Authors
I thank the authors for the corrections made.
Reviewer 2 Report
Comments and Suggestions for Authors
Dear Authors,
I appreciate your careful revisions and your attention to all of my previous comments.